# The nucleolar protein NIFK promotes cancer progression via CK1α/β-catenin in metastasis and Ki-67-dependent cell proliferation

Tsung-Chieh Lin[1†], Chia-Yi Su[1†], Pei-Yu Wu[2], Tsung-Ching Lai[1], Wen-An Pan[2,3], Yi-Hua Jan[1], Yu-Chang Chang[1], Chi-Tai Yeh[4], Chi-Long Chen[5,6], Luo-Ping Ger[7], Hong-Tai Chang[8,9], Chih-Jen Yang[10], Ming-Shyan Huang[10], Yu-Peng Liu[11,12], Yuan-Feng Lin[13], John Y-J Shyy[14], Ming-Daw Tsai[1,2*], Michael Hsiao[1*]

[1]Genomics Research Center, Academia Sinica, Taipei, Taiwan; [2]Institute of Biological Chemistry, Academia Sinica, Taipei, Taiwan; [3]Institute of Biochemical Sciences, National Taiwan University, Taipei, Taiwan; [4]Department of Medical Research and Education, Taipei Medical University-Shuang Ho Hospital, New Taipei City, Taiwan; [5]Department of Pathology, Taipei Medical University Hospital, Taipei Medical University, Taipei, Taiwan; [6]Department of Pathology, College of Medicine, Taipei Medical University, Taipei, Taiwan; [7]Department of Medical Education and Research, Kaohsiung Veterans General, Kaohsiung, Taiwan; [8]Department of Surgery, Kaohsiung Veterans General, Kaohsiung, Taiwan; [9]Department of Emergency Medicine, Kaohsiung Veterans General, Kaohsiung, Taiwan; [10]Department of Internal Medicine, Kaohsiung Medical University Hospital, Kaohsiung, Taiwan; [11]Graduate Institute of Clinical Medicine, Kaohsiung Medical University, Kaohsiung, Taiwan; [12]Center for Infectious Disease and Cancer Research, Kaohsiung Medical University, Kaohsiung, Taiwan; [13]Graduate Institute of Clinical Medicine, College of Medicine, Taipei Medical University, Taipei, Taiwan; [14]Department of Medicine, University of California, San Diego, San Diego, United States

*For correspondence: mdtsai@ gate.sinica.edu.tw (MDT); mhsiao@gate.sinica.edu.tw (MH)

[†]These authors contributed equally to this work

Competing interests: The authors declare that no competing interests exist.

**Abstract** Nucleolar protein interacting with the FHA domain of pKi-67 (NIFK) is a Ki-67-interacting protein. However, its precise function in cancer remains largely uninvestigated. Here we show the clinical significance and metastatic mechanism of NIFK in lung cancer. NIFK expression is clinically associated with poor prognosis and metastasis. Furthermore, NIFK enhances Ki-67-dependent proliferation, and promotes migration, invasion *in vitro* and metastasis *in vivo* via downregulation of casein kinase 1α (CK1α), a suppressor of pro-metastatic TCF4/β-catenin signaling. Inversely, CK1α is upregulated upon NIFK knockdown. The silencing of CK1α expression in NIFK-silenced cells restores TCF4/β-catenin transcriptional activity, cell migration, and metastasis. Furthermore, RUNX1 is identified as a transcription factor of *CSNK1A1* (CK1α) that is negatively regulated by NIFK. Our results demonstrate the prognostic value of NIFK, and suggest that NIFK is required for lung cancer progression via the RUNX1-dependent CK1α repression, which activates TCF4/β-catenin signaling in metastasis and the Ki-67-dependent regulation in cell proliferation.

**eLife digest** Cancer cells can rapidly divide to form a tumor. Small groups of cells can leave the tumor to migrate to other sites in the body, and it is these "secondary" tumors that are often responsible for the death of cancer patients. Many proteins influence how and when cells divide and migrate. One such protein called Ki67 is only produced when cells are dividing and it is often used in the clinic as a marker to indicate whether cells have become cancerous. However, it is not clear how Ki67 regulates the progression of cancer.

Ki67 interacts with another protein called NIFK, and Lin, Su et al. have now investigated the role of NIFK in cancer. First, publicly available data on the levels of proteins in tumor samples from cancer patients were analyzed. This revealed that, in several different types of cancer, tumors that produced more NIFK were more likely to spread to other parts of the body than tumors that produced smaller amounts of NIFK.

Next, Lin, Su et al carried out experiments using human lung cancer cells. This revealed that cells that produced larger amounts of NIFK were more likely to migrate, while cells with lower levels of NIFK divided and migrated less often. Further experiments showed that NIFK increases the activity of genes that are involved in cell migration. NIFK achieves this by reducing the production of a protein that inhibits the activity of another protein called β-catenin.

Lin, Su et al.'s findings reveal a new role for NIFK in promoting the development of cancer. A future challenge is to find out whether chemicals that inhibit NIFK could be used in the treatment of lung cancer.

## Introduction

The Ki-67 protein plays a specific but enigmatic role in cancer. Since its discovery as an antigen for a monoclonal antibody against a Hodgkin lymphoma cell line more than 30 years ago (*Gerdes et al., 1983*; *Gerdes et al., 1984*), Ki-67 serves as the most useful biomarker of cancer in clinical practice (*Brown and Gatter, 2002*; *Yerushalmi et al., 2010*; *Martin et al., 2004*). The finding that Ki-67 is exclusively expressed in the nucleus during all active cell cycle phases (G1, S, G2, and mitosis) in actively proliferating cells but not in quiescent cells renders it a reliable marker for proliferation fraction assessment (*Gerdes et al., 1984*). In many cancer types including breast (*Luporsi et al., 2012*), lung (*Jakobsen and Sorensen, 2013*), brain (*Abry et al., 2010*), and prostate cancer (*Fisher et al., 2013*), the Ki-67 expression level represents a proliferation index, and Ki-67 overexpression predicts poor prognosis. Moreover, Ki-67 has also been used to determine cancer treatment strategies (*Dowsett et al., 2011*). Ki-67 silencing was recently reported to elicit the disappearance of perichromosomal layers in mitosis and the smaller nuclei of daughter cells leading to the problem in mitosis (*Booth et al., 2014*). However, although Ki-67 is known to play an important role in cell proliferation, the mechanism by which Ki-67 regulates cancer progression remains unclear.

As a nuclear protein, Ki-67 interacts with other nuclear proteins (*Sueishi et al., 2000*; *Takagi et al., 2001*). Increasing research focused on the interaction between Ki-67 and other nuclear proteins has paved the way for the elucidation of the mechanism by which Ki-67 regulates cancer progression. Nucleolar protein interacting with the forkhead-associated (FHA) domain of pKi-67 (NIFK) is a nucleolar and cytoplasmic protein that interacts with Ki-67 (*Takagi et al., 2001*; *Li et al., 2004*; *Byeon et al., 2005*). NIFK binds to the FHA domain of Ki-67 by two key regulators of mitosis: cyclin-dependent kinase 1 (CDK1) and glycogen synthase kinase GSK-3 (*Byeon et al., 2005*), and the NIFK sequential phosphorylation at Thr238 and Thr234 is required for the Ki-67 interaction (*Byeon et al., 2005*). During mitosis, both NIFK and Ki-67 are recruited to the chromosome periphery, and the localization of NIFK at the peripheral of mitotic chromosome is disrupted in the absence of Ki-67 (*Booth et al., 2014*; *Van Hooser et al., 2005*). This critical role of the NIFK-Ki-67 interaction in regulating mitosis renders NIFK as a promising target of cancer research. However, to date, studies focused on the importance of NIFK in cancer are lacking, and no clear evidence has emerged to elucidate whether and how NIFK regulates cancer progression.

The Wnt/β-catenin signaling pathway is critical during tumorigenesis and metastasis in various types of cancer (*Polakis, 2000*; *Sinnberg et al., 2010*). Stabilization and nuclear import of β-catenin

activates downstream transcriptional targets, *MMP7*, *MYC*, *TCF4*, *CCND1* and *CD44* which are related to cell cycle, differentiation and metastasis regulation (*Schwartz et al., 2003*; *Dey et al., 2013*; *Cho et al., 2011*). β-catenin is constantly synthesized but is normally controlled at restricted low concentration by proteasome-mediated degradation. Degradation of β-catenin is shown to be regulated via sequential phosphorylation by casein kinase 1α (CK1α) first, and then by GSK-3, which facilitates the formation of the destruction complex (*Hernandez et al., 2012*; *Li et al., 2012*). CK1 family members including CK1α are constitutively active in cells (*Price MA, 2006*). Therefore, CK1α function is determined by its intracellular level. However, the mechanism of CK1α expression regulation in tumors, especially in lung cancer remains obscure.

In this study, we aimed to characterize the role of NIFK, an important Ki-67 binding partner, in cancer progression. The significant association between NIFK and Ki-67 expression in approximately 20 cancer types based on samples from over 7000 patients in a public database confirmed the importance of NIFK in cancer. We focused our study on lung cancer due to the strongest prognostic value of NIFK for lung cancer. Surprisingly, our results revealed NIFK significantly promotes cancer migration and invasion *in vitro* and tumor metastasis *in vivo* in addition to its ability to regulate cancer proliferation. Furthermore, we demonstrated that NIFK modulates lung cancer metastasis by regulating TCF4/β-catenin signaling via the alternation of Casein kinase 1α (CK1α) expression. Our study indicates that NIFK expression promotes cancer metastasis and proliferation leading to poor clinical outcomes; thus, NIFK may represent a prognostic indicator and a promising therapeutic target for lung cancer patients.

## Results

### NIFK expression is most concurrently elevated with Ki67 in lung cancer and lung cancer patients displaying high NIFK level exhibit frequent lymph node and distant metastasis

Due to the well-known characteristics of NIFK as a Ki-67-interacting protein, we first analyzed the expression level of NIFK and Ki-67 based on a public database. Using The UCSC cancer genomics browser web resource, 16 cancer types from the TCGA pan-cancer cohort were analyzed. A significantly positive correlation between *MKI67IP* (NIFK) and *MKI67* (Ki-67) was observed in almost all cancer types (*Figure 1A*). High *MKI67IP* expression was observed in lung, colorectal, breast, uterine, bladder, head and neck, melanoma, cervical, and ovarian cancer. In these high *MKI67IP*-expressing cancer types, the most significant positive correlation between *MKI67IP* and *MKI67* expression was detected in lung cancer (ρ = 0.488, p<0.001). Based on the heat map, we also observed that the normal tissue group tended to display low *MKI67IP* expression. IHC analysis revealed significantly higher expression of NIFK in the samples from our patient cohort than in the paired normal tissue for lung and colorectal cancer but not breast cancer (*Figure 1B*). To identify the cancer types in which NIFK exerts the most significant impact on cancer progression, we examined the prognostic value of NIFK for various cancer types using the PrognoScan database. High *MKI67IP* expression was associated with poor survival in several cancer types, including lung, breast, and blood cancer (*Figure 1C*). By ranking the hazard ratios from the Cox proportional hazards survival model, we determined that high *MKI67IP* expression corresponded to the highest hazard ratio in lung cancer patients (hazard ratio = 4.71, Cox p value = 0.000308). Based on clinicopathological analysis of lung cancer, the patients displaying high NIFK protein expression exhibited more frequent nodal involvement (p = 0.032) and distant metastasis (p = 0.036), as well as a higher pathological stage (p = 0.059) (*Figure 1D*). Similar results were observed in a lung cancer cohort from the TCGA database (*Figure 1—figure supplement 1*). According to the above results, NIFK displayed the greatest clinical significance for lung cancer and may be associated with lung cancer progression by regulating tumor metastasis.

### NIFK promotes the migration and invasion of lung cancer cells *in vitro*

Based on the prognostic significance of NIFK and the strong correlation between NIFK and Ki-67 expression in lung cancer, we further investigated the functional role of NIFK *in vitro* and *in vivo*. In 12 lung cancer cell lines, the endogenous NIFK levels were normalized to those of the normal lung cell line Beas2B (*Figure 2A*, Top), and the relative NIFK levels were statistically analyzed (*Figure 2A*,

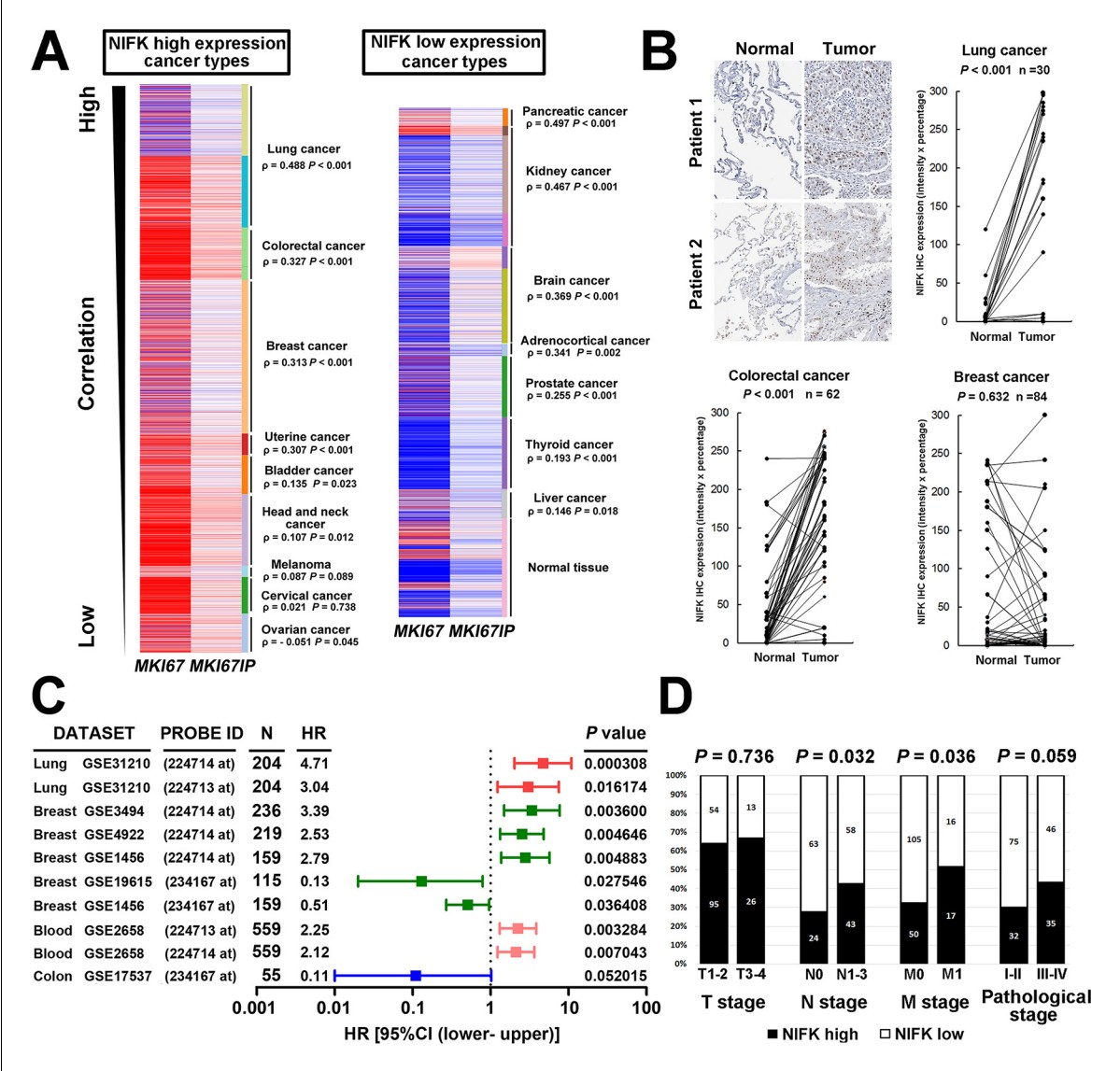

**Figure 1.** NIFK expression is most concurrently elevated with Ki67 in lung cancer and lung cancer patients displaying high NIFK level exhibit frequent lymph node and distant metastasis. (A) In the TCGA pan-cancer cohort, significantly positive correlations between *MKI67IP* (NIFK) and *MKI67* (Ki-67) RNA expression were observed in almost all cancer types. Among the cancer types that displayed high *MKI67IP* expression, lung cancer exhibited the strongest correlation between *MKI67IP* and *MKI67* expression. Red color in heat map represents genes with high expression. Blue color in heat map represents gene with low expression. (B) Based on IHC analysis, significant overexpression of NIFK compared with paired normal tissue was observed in lung cancer (upper) and colorectal cancer (lower left) but not in breast cancer (lower right). (C) Forest plot comparison of the hazard ratio of *MKI67IP* (NIFK) overexpression in patients with various cancer types revealed that lung cancer displayed the strongest impact of NIFK RNA expression on survival. (D) Lung cancer patients displaying NIFK overexpression exhibited more frequent lymph node and distant metastasis and a higher pathological stage.

The following figure supplement is available for figure 1:

**Figure supplement 1.** *MKI67IP* is overexpressed in tumors and its expression correlates with the pathological TNM stage in lung cancer.

Bottom). Then, we stably overexpressed NIFK in A549 and PC13 cells, which exhibit low NIFK expression, via lentiviral infection (*Figure 2B*). A significant increase in cell migration after NIFK over-expression was observed in the wound-healing assay (p<0.001 for 4 MOI, *Figure 2C*). Increased cell migration and invasion upon NIFK overexpression were confirmed by the transwell assay with short incubation time (*Figure 2—figure supplement 1*). In addition, higher endogenous NIFK levels were

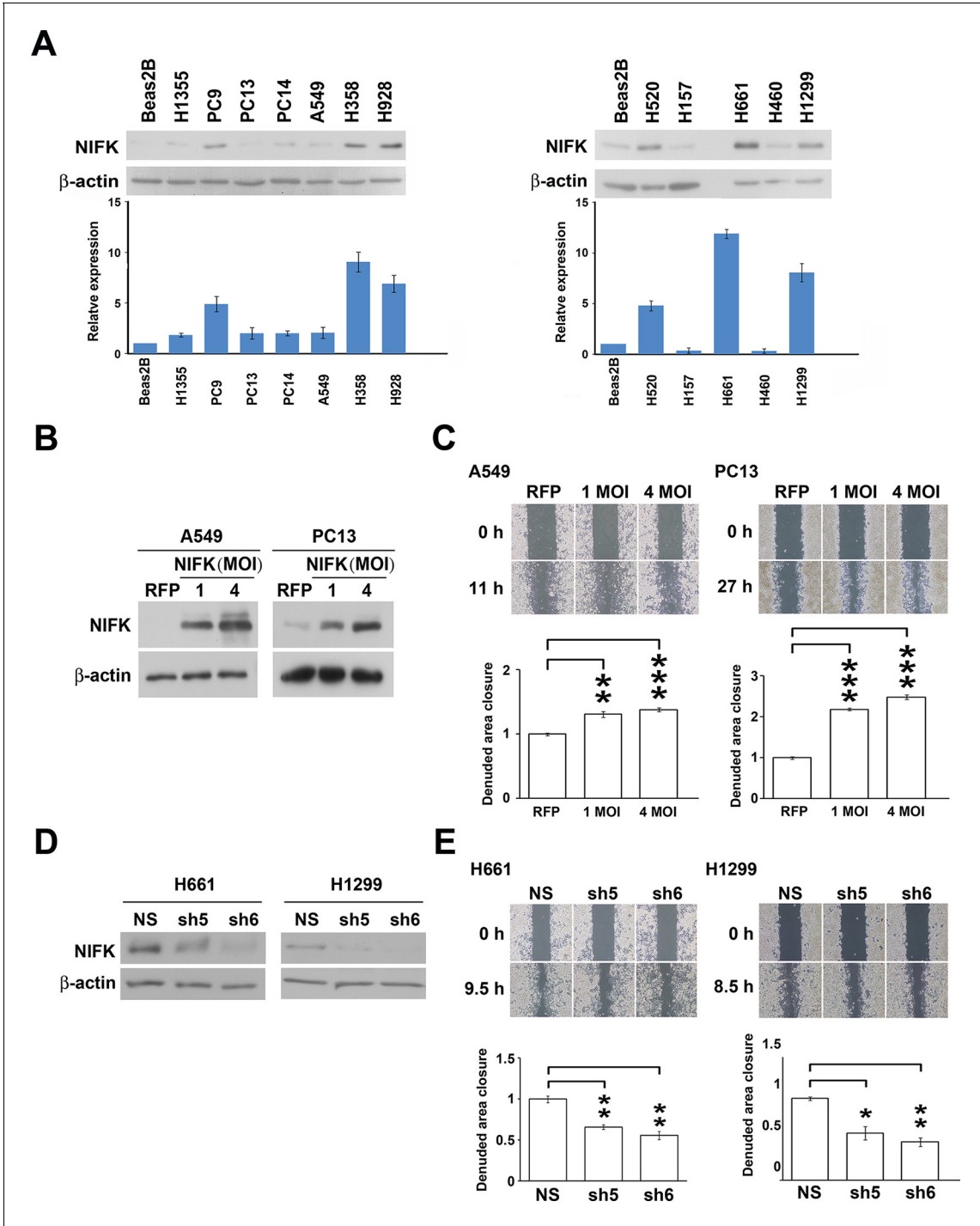

**Figure 2.** NIFK promotes the migration of lung cancer cells *in vitro*. (A) The endogenous expression levels of NIFK in lung adenocarcinoma (Left), squamous and large cell lung cancer cell lines (Right). The relative expression levels were normalized to those of normal Beas2B lung cells, and the average expression levels are presented. (B) The relative NIFK levels in A549 and PC13 cells after overexpression of NIFK via lentiviral infection. (C) The migratory capacity of NIFK-overexpressing A549 and PC13 cells was assessed using a wound-healing assay. The exposed area was measured after the indicated incubation period and was normalized to that of the 0-h control. (D) The NIFK knockdown efficiencies in the lentivirus-based shRNA clones sh5 and sh6, corresponding to H661 and H1299 cells, respectively. NS, non-silenced control. (E) H661 and H1299 cell migration after NIFK knockdown was evaluated at the indicated time points.

The following figure supplement is available for figure 2:

**Figure supplement 1.** The overexpression of NIFK promotes cell migration, invasion *in vitro* and metastasis *in vivo*.

detected in H661 and H1299 cells, which were derived from metastatic sites in lung cancer patients (ATCC) and are considered to be invasive (*Figure 2A*). Therefore, we established stable NIFK-silenced H661 and H1299 clones using lentivirus-based shRNA-mediated knockdown (*Figure 2D*). Cell migration was inhibited after NIFK knockdown, especially in clone sh6 (p<0.01, *Figure 2E*). *The in vitro* results revealed that NIFK could promote cell migration and invasion.

## NIFK facilitates cancer cell metastasis *in vivo* and is associated with poor survival of lung cancer cohorts

Additionally, *in vivo* animal model experiments were performed to examine whether NIFK promotes tumor metastasis. A549 cells overexpressing NIFK were injected into NSG mice via the tail vein (*Figure 3A*). Increased lung metastasis was observed in the NIFK-overexpressing group both grossly and microscopically (p = 0.0127, *Figure 3B–C* and *Figure 3—figure supplement 1*). Similarly, significantly reduced numbers of metastatic lung nodules were observed in the NIFK-silenced H661 cell-injected group (p = 0.0054) and in the NIFK-silenced H1299 cell-injected group (p = 0.0169) compared with the corresponding non-silenced (NS) control cell-injected groups (*Figure 3D–E*). Representative images of HE staining and IHC staining for NIFK are presented in *Figure 3F*. In our patient cohort, high NIFK IHC expression significantly correlated with poor overall survival in both the Taiwanese (p = 0.018) and Korean (p = 0.041) lung cancer cohorts (*Figure 3G–H*). These results including clinical data suggest a regulatory role of NIFK in lung cancer progression, especially in tumor metastasis.

## NIFK promotes cell proliferation via the poor prognosis marker Ki-67

Due to the previously characterized NIFK as a Ki-67-interacting protein, we therefore studied the regulatory role of NIFK in cell proliferation. NIFK downregulation in H661 and H1299 cells decreased the cancer proliferation at 96 hr *in vitro* (*Figure 4A*). Furthermore, H1299 cell with NIFK silencing displayed the reduced tumor weight and volume *in vivo* p<0.01, *Figure 4B–C*). The NIFK-regulated cell proliferation was Ki-67-dependent that Ki-67 silencing decreased cell growth in both A549 and PC13 cells (*Figure 4D*). In addition, the effect of NIFK Ki-FHA binding, RNA recognition motif (RRM) truncation and T234A/T238A point mutation on cell proliferation was compared with ectopic expression of wild type (WT) NIFK (*Figure 4E–F*). Based on *Figure 4F*, both RRM and Ki-FHA binding domain were involved in tumor proliferation, but the latter was more significant. In lung cancer cohort, we observed positive correlation of high NIFK/Ki-67 with poor survival at protein level (*Figure 4G*, p = 0.004) and significant expression coefficient (*Figure 4H*, p<0.001). The results potentially indicated the requirement of Ki-67 in NIFK-increased cell proliferation and the significance of NIFK and Ki-67 in lung cancer progression.

## Knowledge-based analysis of the microarray data reveals that NIFK regulate CK1α and Wnt signaling

The importance of NIFK in clinical patients motivated us to explore the molecular mechanism underlying NIFK-mediated lung cancer metastasis. Knocking down Ki-67 expression by shRNA1 in NIFK-overexpressing PC13 cells and H1299 cells did not impact NIFK-induced cell migration or invasion (*Figure 5A–D*). Therefore, we performed microarray analysis of PC13 cells overexpressing NIFK. The signature of genes displaying a≥1.5-fold change in expression upon NIFK overexpression was subjected to Ingenuity Pathway Analysis (IPA) and MetaCore database analysis (*Supplementary file 1A*). Based on the MetaCore enrichment analysis, Wnt signaling, a previously identified pivotal pathway in regulating cancer metastasis was most strongly affected by NIFK overexpression (*Figure 5E–F*). The levels of CK1α and the adenomatous polyposis coli (APC)-AXIN-CK1α-CTNNβ-GSK3β complex were decreased following NIFK overexpression, indicating that CK1α is a key molecule in the NIFK-regulated Wnt signaling pathway (*Figure 5G–H*, *Figure 5—figure supplement 1*). Based on these bioinformatics results, we hypothesized that NIFK overexpression downregulates CK1α, thereby decreasing the levels of the APC-AXIN-CK1α-CTNNβ-GSK3β complex, which in turn increases intracellular β-catenin protein stability. The increased nuclear fraction of β-catenin was observed upon NIFK overexpression (*Figure 5I*) as well as the reduced β-catenin phosphorylation (*Figure 5—figure supplement 2*). Furthermore, we determined whether TCF/β-catenin transcriptional activity is affected by the NIFK/CK1α axis. The results of TCF/β-catenin reporter assay

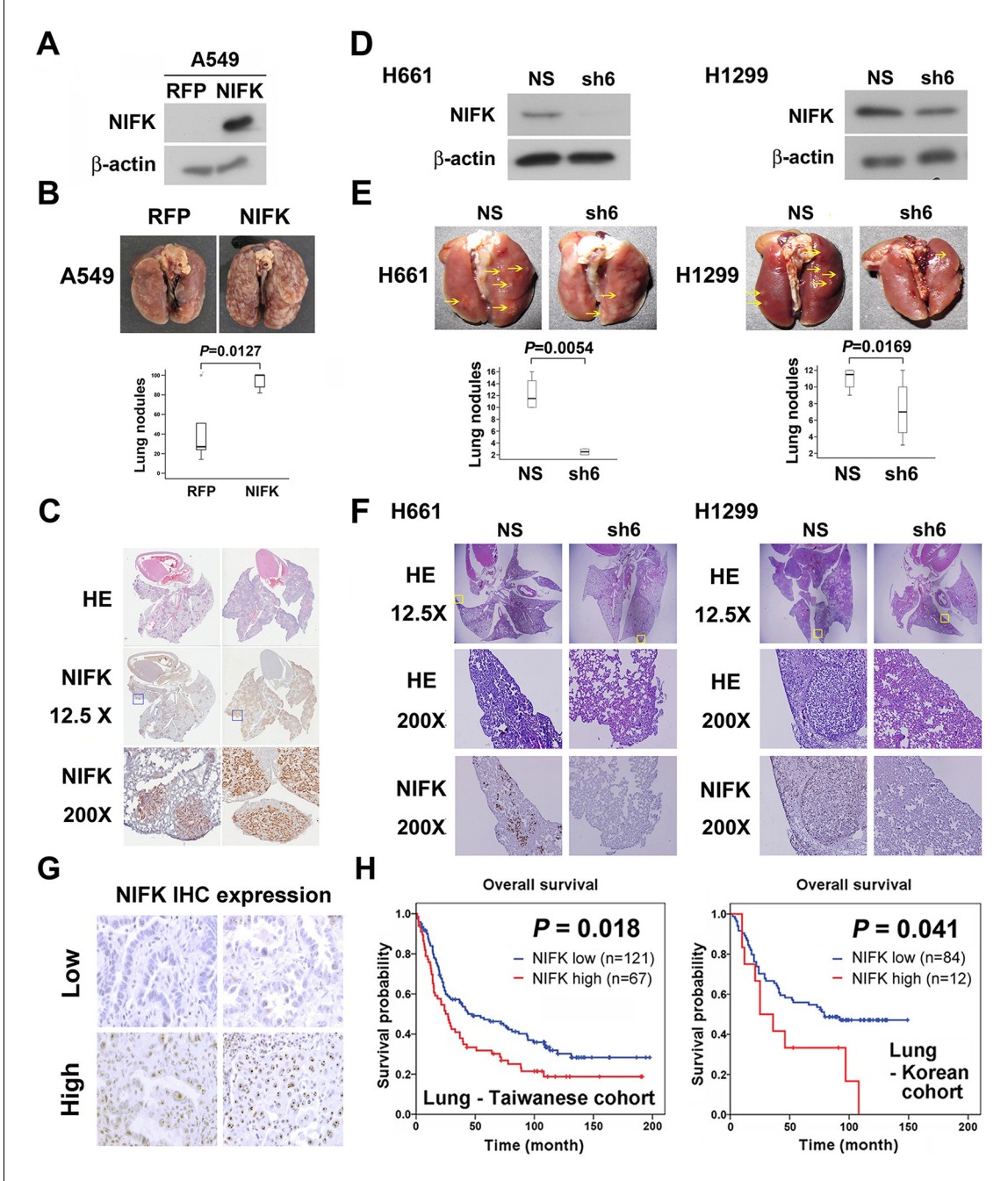

**Figure 3.** NIFK facilitates cancer cell metastasis *in vivo* and is associated with poor survival of lung cancer cohorts. (A) The level of NIFK overexpression following infection of A549 cells with 4 MOI virus. (B) The indicated cells were injected into NSG mice via the tail vein. Surface lung nodules were statistically quantified. N=5 per group. (C) Representative images of surface lung nodules with HE staining and IHC staining for NIFK are presented for the RFP- or NIFK-overexpressing cell-injected groups. (D) The NIFK knockdown efficiency in H661 and H1299 cells. (E) The cells were injected via the tail vein of the mice. Top, representative images of lung metastasis of the indicated H661 and H1299 cells. Bottom, statistical quantification of lung metastatic nodules in the indicated groups. (F) Top, representative images of lung HE staining. Middle, images of HE staining in the indicated areas. Bottom, IHC staining for NIFK in the indicated areas. (G) Representative images of IHC staining for NIFK. (H) Kaplan-Meier survival analysis revealed that high NIFK IHC expression correlates with poor prognosis in lung cancer.

The following figure supplement is available for figure 3:

**Figure supplement 1.** Cells overexpressing NIFK or RFP were injected via the tail vein.

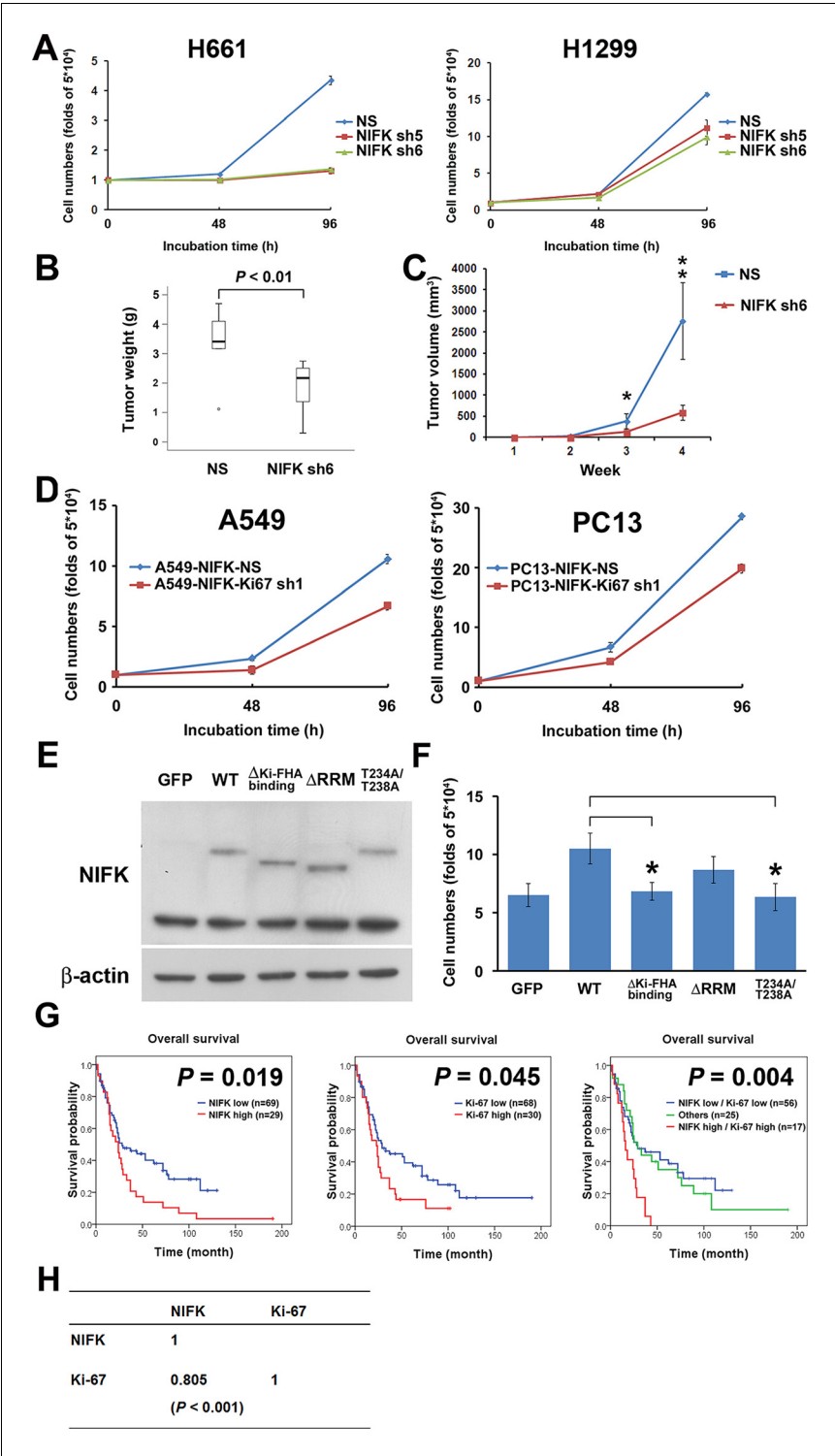

**Figure 4.** NIFK promotes cell proliferation via the poor prognosis marker Ki-67. (**A**) H661 and H1299 cells were seeded at a density 5×10⁴ cells/well. The number of cells was counted at 48 and 96 hr via the Trypan blue exclusion assay. (**B**) Box plot represented tumor weight of H1299 cell NS clone and NIFK-silenced clone at 4 weeks after injection. (**C**) Growth curves of tumor volume in indicated groups. (**D**) Cell numbers of NIFK overexpressing A549 and PC13 cells upon Ki-67 silence. (**E**) Overexpression of GFP-tagged wild type (WT), truncated and point mutated NIFK in PC13 cells. Cells were transiently transfected via liposome and selected. (**F**) Cell numbers of indicated groups were evaluated at 48 hr after cell seeding at density 5×10⁴ cells/ well in 6 wells plate. (**G**) Kaplan-Meier survival analysis revealed the correlation of NIFK and Ki-67 IHC expression with poor prognosis in lung cancer. (**H**) The correlation of NIFK and Ki-67 IHC expression in lung cancer.

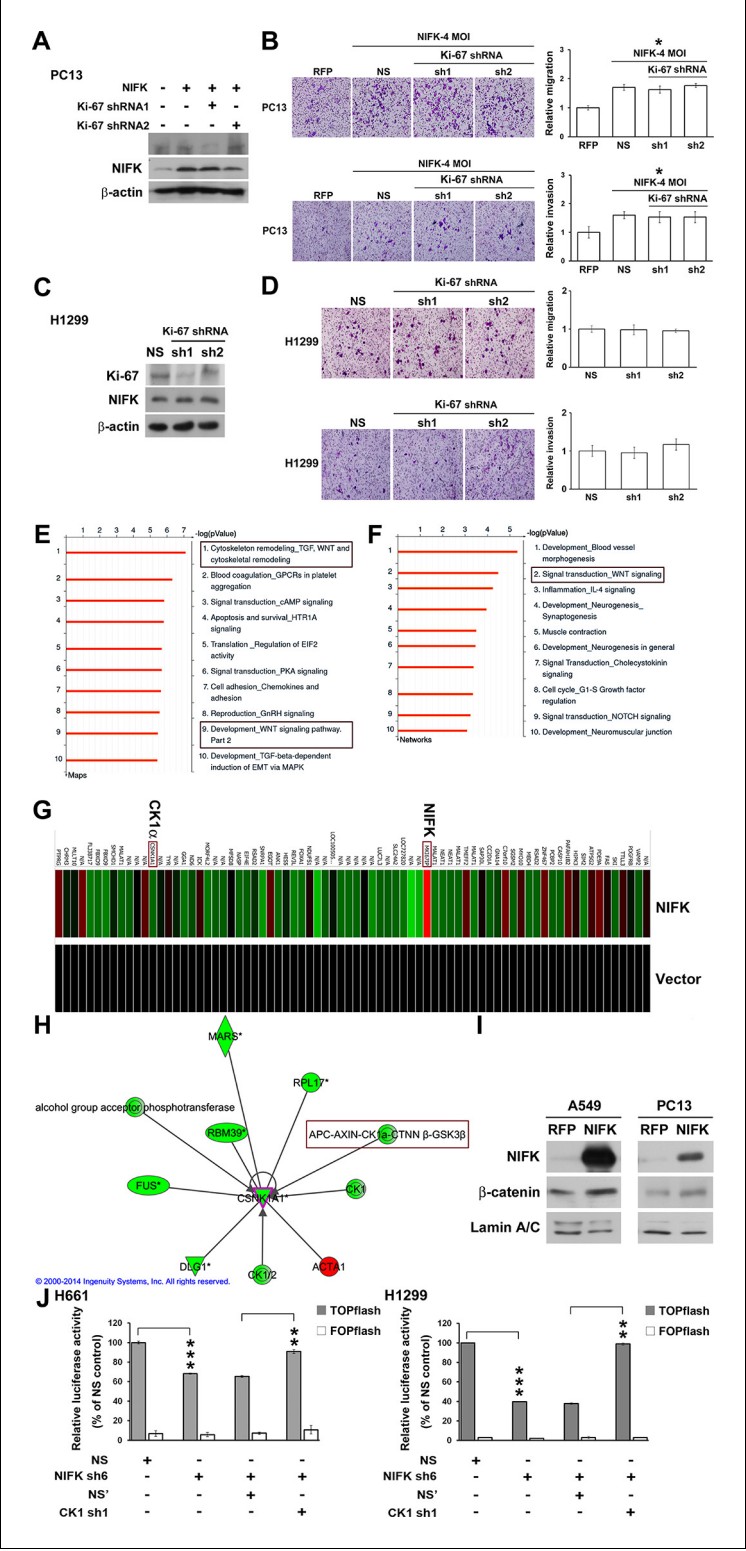

**Figure 5.** Knowledge-based analysis of the microarray data reveals that NIFK regulate CK1$\alpha$ and Wnt signaling. (A) The relative knockdown efficiencies of Ki-67 in (A) PC13 and (C) H1299 cells. Cell migration and invasion were evaluated via the transwell assay using (B) PC13 and (D) H1299 cells. (E) List of the top 10 signaling pathways altered by NIFK overexpression. The microarray data of NIFK overexpression in PC13 cells were analyzed using the MetaCore Maps database. (F) List of the top 10 networks affected by NIFK overexpression. The gene signatures were processed via the MetaCore Networks database. (G) Representative signature of genes displaying a≥1.5-fold

*Figure 5 continued on next page*

*Figure 5 continued*

change in expression due to NIFK overexpression in PC13 cells. The red and green bars represent upregulation and downregulation, respectively. (H) Knowledge-based IPA analysis of the microarray data focusing on CK1α (*CSNK1A1*)-mediated signaling. The red and green circles represent upregulation and downregulation, respectively. (I) The nuclear levels of β-catenin in cells with NIFK overexpression were showed. Nuclear fractions were extracted from A549 and PC13 cells. (J) TCF/LEF transcriptional activity alteration in the indicated H661 and H1299 cells. TOPflash: reporter plasmid with TCF binding sites. FOPflash: reporter plasmid with mutated TCF binding sites.

The following figure supplements are available for figure 5:

**Figure supplement 1.** NIFK down-regulates CK1α, and up-regulates β-catenin level especially in nucleus.

**Figure supplement 2.** Phospho-β-catenin levels are decreased after NIFK overexpression.

**Figure supplement 3.** NIFK regulates downstream transcriptional targets of TCF4/LEF via CK1α.

**Figure supplement 4.** NIFK regulates downstream transcriptional targets of β-catenin.

---

revealed the decreased luciferase activity in the NIFK-silenced cells. Additional knockdown of CK1α restored TCF/β-catenin transcriptional activity in both NIFK-silenced H661 and H1299 cells (p<0.01, *Figure 5J*). In addition, the downstream transcriptional targets of TCF/β-catenin, including *TCF4*, *CD44*, *CCND1* and *MMP7*, were regulated by the NIFK-CK1α axis (*Figure 5—figure supplement 3*). The upregulated downstream transcriptional targets were further confirmed by another microarray analysis of PC13 cells with lentiviral-based stable NIFK overexpression (*Figure 5—figure supplement 4* and *Supplementary file 1B*, labeled in red). These data indicated that NIFK might regulate TCF/β-catenin-mediated transcriptional events via CK1α.

## RUNX1 may act as a transcription factor of CK1α to participate in the NIFK-induced downregulation of CK1α mRNA expression

Knocking down NIFK expression elicited CK1α upregulation, which decreased the downstream inhibitory target β-catenin in the H661 and H1299 cell lines (*Figure 6A*, Left). In addition, NIFK over-expression using a lentivirus decreased CK1α expression and increased β-catenin level in A549 and PC13 cells (*Figure 6A*, Right). A similar result was observed following transient overexpression of NIFK (*Figure 6—figure supplement 1*). To investigate the potential mechanism underlying the NIFK-mediated regulation of *CSNK1A1* (CK1α) transcription, we performed a search using the TFSEARCH website to identify putative transcription factors that may bind to the promoter. Several potential transcription factors were identified according to the matched consensus binding sequences of each candidate transcription factor to the *CSNK1A1* (Ensembl:ENSG00000113712) promoter region (*Figure 6B*). RUNX1 and SRY scored 100 with respect to the matched sequences. The promoter region from -969 to -127 bp was found to be important for transcriptional activation, indicating the potential involvement of SRY and RUNX1 (*Figure 6C*). Thus, we determined the involvement of these transcription factors in regulating the expression level of CK1α. Silencing of RUNX1 and SRY expression in H1299 cells using shRNA decreased the CK1α expression levels (*Figure 6D*), whereas no change in CK1α expression was detected following knockdown of another putative transcription factor, CdxA (*Figure 6—figure supplement 2*). Furthermore, The ChIP assay was performed on NS H1299 cells and the NIFK-silenced H1299 cell clone sh6 to explore whether the binding of the predicted transcription factors to the promoter region is regulated by NIFK. As shown in *Figure 6E*, increased promoter binding of RUNX1 was observed in the NIFK-silenced H1299 cell clone sh6 compared with the NS control clone (Top panel). In a complementary experiment, the overexpression of NIFK in PC13 cells decreased the promoter binding of RUNX1 (*Figure 6E*, Bottom panel). However, this regulation was not significantly observed for SRY based on the ChIP experiment (*Figure 6E*). Thus, we ruled out the involvement of SRY in the NIFK/CK1α axis. To further determine whether RUNX1 affects CK1 promoter activity, we performed the reporter assay. Our

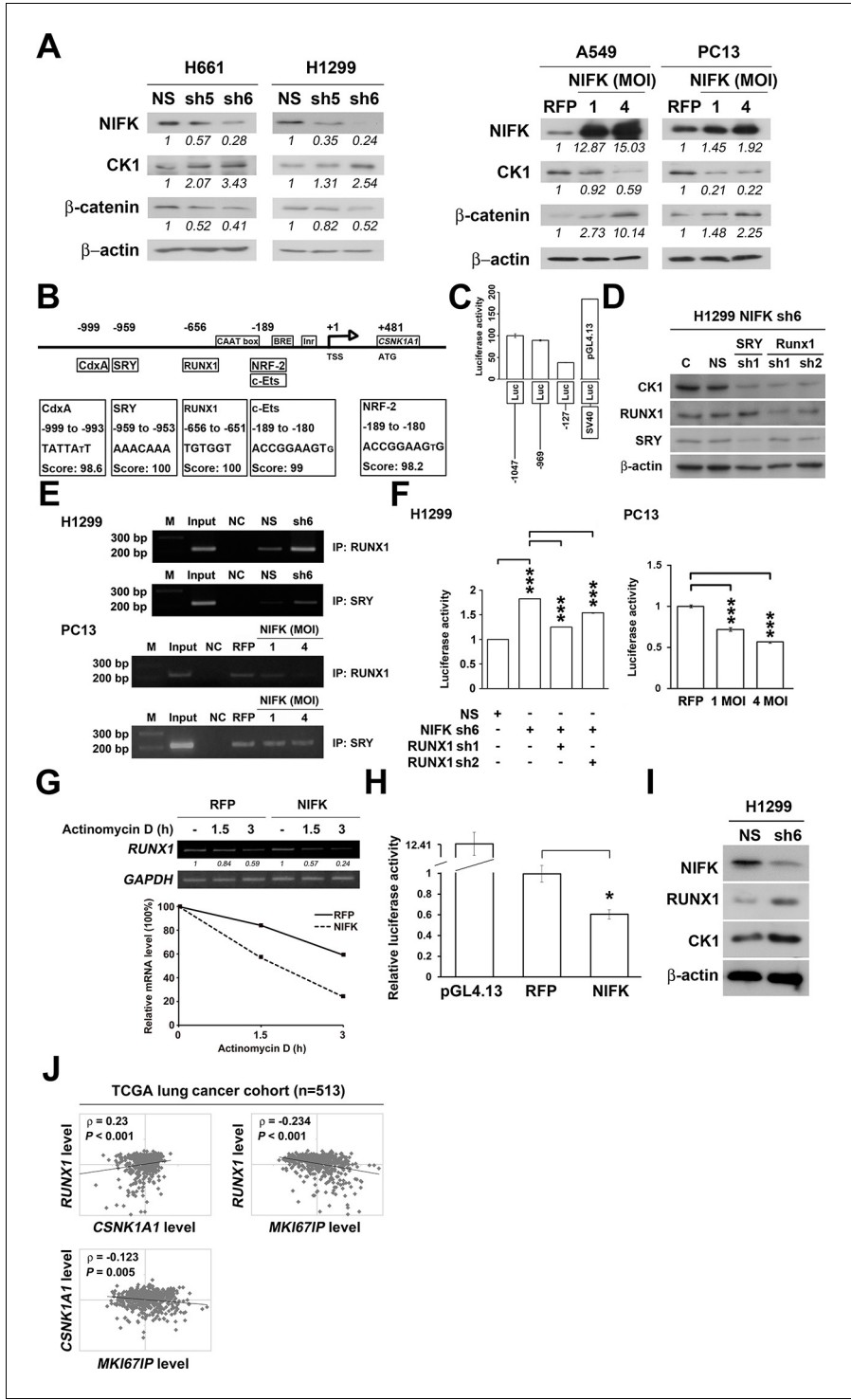

**Figure 6.** NIFK regulates CK1α expression via the destabilization of transcription factor RUNX1. (**A**) Left, the levels of CK1α and β-catenin in H661 and H1299 cells after NIFK knockdown. Right, the levels of the indicated molecules in A549 and PC13 cells upon NIFK overexpression. (**B**) Identification of *cis*-regulatory elements within the *CSNK1A1* (Ensembl:ENSG00000113712) promoter region. The locations of the consensus binding sites relative to the transcription start site (TSS) are presented below the indicated transcription factors. The scores were calculated by the TFSEARCH website according to the matched sequences. BRE, B recognition element. Inr, initiator element. (**C**) The CK1 promoter region from -969 to -127 bp is important for transcriptional activation. The relative luciferase activity was measured 48 hr post-transfection with a reporter plasmid containing the CK1 promoter or the indicated deletion mutant. (**D**) The expression of the indicated molecules after the knockdown of

*Figure 6 continued*
the transcription factors SRY and RUNX1 in the NIFK-silenced H1299 cell clone sh6. (E) ChIP was performed on H1299 (Top) and PC13 cells (Bottom) using antibodies against RUNX1 and SRY. M, DNA marker. NC, negative beads control. (F) A CK1 promoter reporter assay was performed on NIFK-silenced H1299 cells (Left) and in NIFK-overexpressing PC13 cells (Right). The cells were lysed 48 hr after reporter plasmid transfection. The relative luciferase activity was normalized to the number of cells and was quantified. (G) A549 cells were transiently transfected with RFP control or NIFK by lipofection. After 48 h, cells were treated with actinomycin D (5 μg/ml) for indicated time points. Relative RUNX1 mRNA levels were analyzed by RT-PCR. (H) RUNX1 promoter activity was measured after 48 hr of transfection by liposome. (I) NIFK, CK1α and RUNX1 protein levels were analyzed in H1299 cells upon NIFK silencing. (J) Correlation between the expression levels of the indicated molecules in the lung cancer cohort. The microarray data were retrieved from the TCGA database (genomic_TCGA_LUAD_exp_HiSeqV2_percentile_clinical).
The following figure supplements are available for figure 6:

**Figure supplement 1.** A549 and PC13 cells were transfected with Flag-tagged NIFK or a vector control for 48 hr.
**Figure supplement 2.** RUNX1 and SRY are potential transcription factors of CK1α.
**Figure supplement 3.** NIFK overexpression leads to RUNX1 mRNA instability.
**Figure supplement 4.** NIFK decreases RUNX1 at RNA level.

results revealed increased CK1 reporter activity after NIFK knockdown in H1299 cells, which was decreased by RUNX1 knockdown (p<0.001, Left, *Figure 6F*). In addition, the relative luciferase activity was dose-dependently decreased by NIFK overexpression in PC13 cells (p<0.001, Right, *Figure 6F*). Furthermore, *RUNX1* mRNA level was found to be destabilized upon NIFK overexpression (*Figure 6G* and *Figure 6—figure supplement 3*), and the promoter activity was decreased by NIFK (*Figure 6H*). Inversely, RUNX1 was increased upon NIFK silencing (*Figure 6I*). In addition, the NIFK-mediated RUNX1 repression might be relevant to its FHA and RRM domain (*Figure 6—figure supplement 4*). However, the binding of NIFK with *RUNX1* mRNA and with RUNX1 protein were not detected (RNA binding protein IP and protein IP data not shown). Therefore, the mechanism needs further to be investigated. In the lung cancer cohort, the NIFK and RUNX1 or CK1α RNA levels were inversely correlated, whereas the RUNX1 and CK1α RNA expression levels were positively correlated (p<0.001, *Figure 6J*). These results suggested that RUNX1 is a potential transcriptional factor of *CSNK1A1* (CK1α) that is negatively regulated by NIFK to decrease CK1α expression.

## NIFK promotes lung cancer metastasis via CK1α and lung cancer patients with high NIFK/low CK1α represent poor survival rate

We further investigated whether NIFK-induced tumor metastasis is mediated by the CK1α. CK1α expression was reduced in H661 and H1299 cells via lentivirus-based shRNA-mediated knockdown in conjunction with NIFK silencing using sh6 to determine the functional role of CK1α (CK1α sh1-3, *Figure 7A*). Knocking down CK1α expression restored cell migration compared with the NIFK knockdown-only group, although the effects on clones sh1 and sh2 were clearer than those on clone sh3 (*Figure 7B*, *Figure 7—figure supplement 1*). These results suggested that the effect of NIFK on cell migration may be regulated by CK1α. Next, we performed animal experiments to examine whether CK1α is involved in NIFK-regulated tumor metastasis. The formation of surface lung nodules was significantly restored due to CK1α knockdown in the H661 and H1299 sh6 cells (*Figure 7C–D*, upper left). This restoration of tumor metastasis was further confirmed via HE staining of the lung tissue (Lower left, *Figure 7C–D*). The quantifications for the number of metastatic lung nodules in each group are presented (Right, *Figure 7C–D*). These results suggested that NIFK regulates lung cancer metastasis via CK1α. In support of this hypothesis, we observed that high levels of CK1α correlated with a favorable prognosis in lung cancer patients (p = 0.051) (*Table 1* and *Figure 7E*). Furthermore, the combination of NIFK (*MKI67IP*) and CK1α (*CSNK1A1*) may represent a superior prognostic

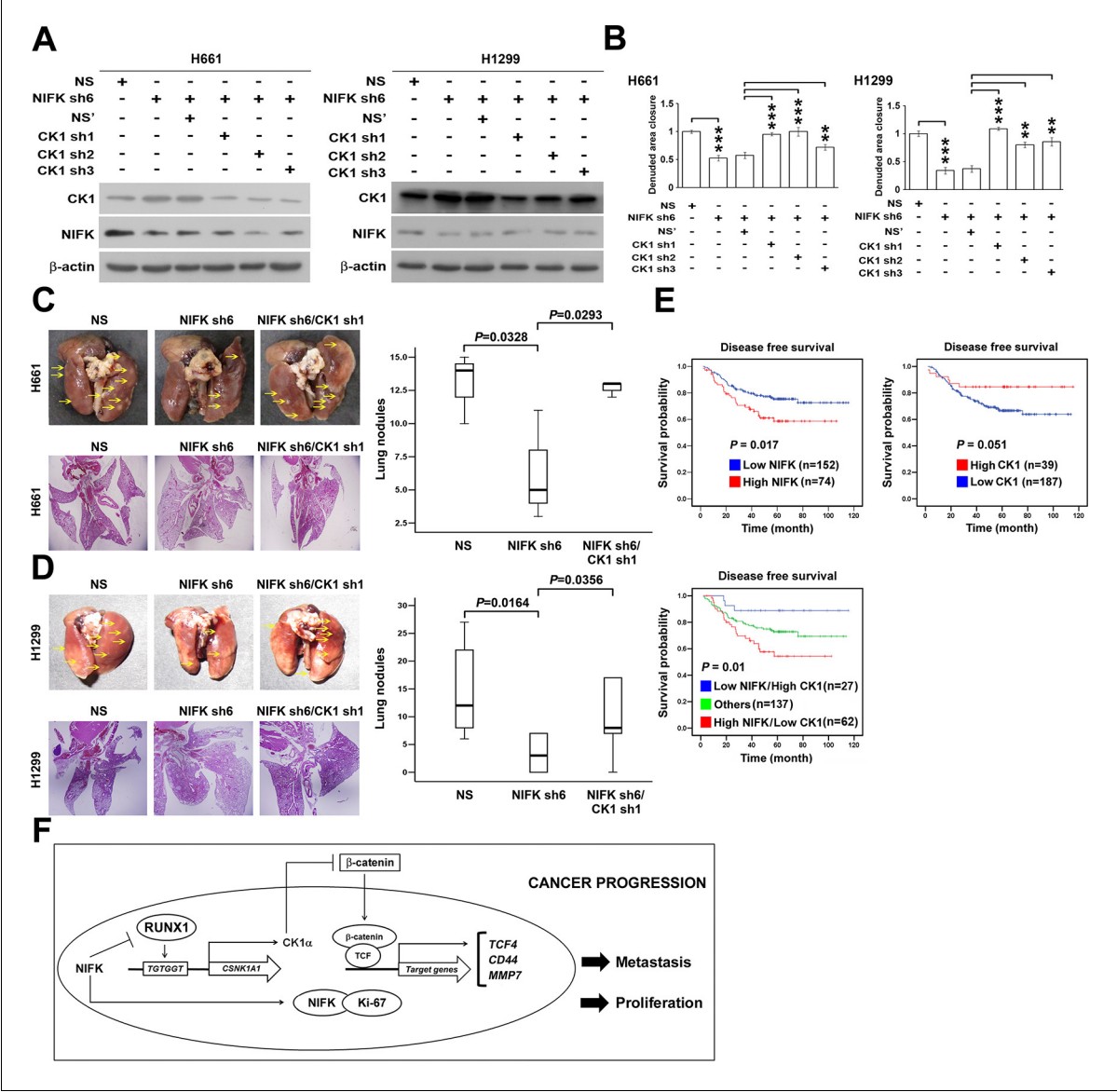

**Figure 7.** NIFK promotes lung cancer metastasis via CK1α and lung cancer patients with high NIFK/low CK1α represent poor survival rate. (**A**) The CK1α knockdown efficiencies of 3 lentivirus-based shRNAs in H661 and H1299 cells. (**B**) Wound-healing assays were performed on the cells in the indicated groups. The quantification of the migration of H661 and H1299 cells is presented. (**C&D**) Animal studies were performed on the indicated groups. Left, representative images of lung metastasis and lung HE staining of NSG mice injected with (**C**) H661 cells or (**D**) H1299 cells. Right, statistical quantification of the number of metastatic nodules in each group is presented. (**E**) Kaplan-Meier plot demonstrating the disease-free survival of 226 lung cancer patients displaying varying (upper left) NIFK (*MKI67IP*) and (upper right) CK1α (*CSNK1A1*) expression levels. (Lower) Kaplan-Meier plot demonstrating the disease-free survival of cases separated into high NIFK/low CK1α and low NIFK/high CK1α groups. The data were retrieved from the microarray analysis of the GSE31210 dataset. (**F**) The model of NIFK-induced activation of TCF/β-catenin transcriptional activity via the Runx-1-dependent downregulation of CK1α in metastasis.

The following figure supplement is available for figure 7:

**Figure supplement 1.** NIFK promotes cancer cell migration via CK1α.

indicator for lung cancer (p = 0.01) (*Figure 7E*). *Figure 7F* represents the model that NIFK promotes lung cancer progression by (1) downregulating CK1α expression through the destabilization of its novel transcription factor, RUNX1, to promote lung cancer metastasis as well as (2) increasing the Ki-67-dependent cell proliferation.

**Table 1.** Cox univariate and multivariate regression analysis of *CSNK1A1* (CK1α) expression for relapse-free survival in GSE31210 lung cancer dataset

| Survival | Variable | Univariate | | Multivariate | |
|---|---|---|---|---|---|
| | | HR (95% CI) | P | HR (95% CI) | P |
| Relapse free (n=246) | CSNK1A1 (high/low) | 2.261 (0.975-5.245) | 0.057 | 1.804 (0.769-4.231) | 0.175 |
| | Age | 1.658 (0.984-2.796) | 0.058 | 1.638 (0.97-2.766) | 0.065 |
| | Sex (female/male) | 1.271 (0.778-2.075) | 0.338 | 0.986 (0.489-1.986) | 0.968 |
| | Smoking habit (never/ever) | 1.333 (0.815-2.178) | 0.252 | 1.19 (0.589-2.402) | 0.628 |
| | Stage (I/II) | 3.163 (1.92-5.21) | 0.001 | 2.912 (1.745-4.862) | 0.001 |

## Discussion

In this study, we demonstrate that NIFK promotes cancer progression by regulating cancer metastasis and proliferation. Lung cancer patients displaying high NIFK expression exhibited poor prognosis and frequent lymph node and distant metastasis. Although Ki-67 has been a research emphasis of cancer, Ki-67-interacting proteins have not been given the attention that they merit and are generally considered to regulate cancer proliferation via their physiological roles in mitosis and the cell cycle (*Booth et al., 2014*; *Byeon et al., 2005*; *Florian and Mayer, 2011*). Hklp2/KIF15, another Ki-67 interacting protein, plays a critical role in the maintenance of spindle bipolarity during cell division and was reported to serve as a breast cancer tumor antigen (*Florian and Mayer, 2011*; *Rath and Kozielski, 2012*; *Scanlan et al., 2001*). NIFK was previously implicated as a c-Myc-responsive gene in breast cancer (*Musgrove et al., 2008*). The significant cancer metastasis-promoting characteristics of NIFK demonstrated by our results provide novel insight into the role of NIFK in cancer. Furthermore, aside from its role as a proliferation marker, Ki-67 has been demonstrated to be associated with metastasis and lymphovascular invasion in cancer patients (*Pollack et al., 2004*; *Inwald et al., 2013*). However, our results imply that Ki-67 may not affect cancer metastasis via its interaction with NIFK.

We showed the NIFK-induced cell proliferation is dependent on Ki-FHA binding motif indicating the requirement of NIFK-Ki-67 interaction in lung cancer proliferation. Ki-67 was reported as a key factor organizing chromosomal periphery during cell mitosis, and silence of Ki-67 might elicit cell death (*Booth et al., 2014*). NIFK interacts with Ki-67 in mitotic phase, and whether the interaction is critical in nuclear organization remains to be explored. Our previous study characterized the role of NIFK in U2OS cell proliferation as well as the requirement of its RRM for rRNA maturation (*Pan et al., 2015*). However, Ki-67 interacting motif of NIFK also plays a role in regulating cell proliferation especially after long-term incubation as compared with the immediate response of RRM deletion in U2OS cell (*Pan et al., 2015*). Both results evidently point on NIFK as a gatekeeper in the maintenance of well-regulated cell cycle. The discrepancy of RRM efficacy in proliferation among two studies might potentially due to the difference in cell type-specific mechanism.

CK1α, a component of the β-catenin destruction complex, is characterized as a negative regulator that blocks Wnt signaling pathway-mediated metastasis (*Clevers, 2006*). In the canonical Wnt signaling pathway, the level of intracellular β-catenin is modulated by proteasomal degradation mediated by destruction complex, which is composed of APC, Axin, CK1α, and GSK3β (*MacDonald et al., 2009*). Given that β-catenin and APC mutation is less commonly observed in lung cancer than in other cancer types such as colon cancer, understanding the mechanism by which these Wnt signaling inhibitors regulate the β-catenin destruction complex is crucial (*Stewart, 2014*). Many studies have demonstrated that the downregulation of the members of the β-catenin destruction complex, including CK1α, is commonly observed in lung cancer cell lines and cancer tissue and correlates with poor prognosis or poor clinicopathological characteristics (*Yang et al., 2013*; *Srivastava et al., 2012*; *Lee et al., 2013*). In our study, we found that NIFK acts as a critical regulator that prevents CK1α-

mediated β-catenin degradation, which in turn leads to cancer metastasis. Therefore, these results highlight NIFK as a novel regulator of the Wnt/β-catenin signaling pathway.

In our study, we also observed the sole downregulation of CK1α upon NIFK overexpression, whereas the expression levels of other components of the destruction complex were not significantly altered. Recently, a negative feedback loop of Wnt signaling activation was reported via the Huwe-1-dependent ubiquitylation of Dishevelled (Dvl) (*de Groot et al., 2014*). Upon Wnt signaling activation, Dvl recruits Axin and GSK3-β away from the destruction complex, thereby inhibiting β-catenin destruction complex formation and, subsequently, β-catenin degradation (*de Groot et al., 2014*; *Stambolic et al., 1996*; *Behrens et al., 1998*). Correspondingly, we detected significant Huwe-1 upregulation based on the microarray data of NIFK overexpression, suggesting the activation of an alternative negative feedback loop via the decrease in Dvl activity (*Supplementary file 1A*). Although this negative feedback might serve to regulate lung cancer progression via GSK-3β, CK1α, as the downstream rate-determining enzyme that sequentially phosphorylates β-catenin prior to GSK3β (*Hernandez et al., 2012*), remains to play a significant role in NIFK regulated β-catenin degradation.

An additional interesting finding in our data is that higher levels of NIFK were observed in p53 loss-of-function mutation H661 cell lines and in p53-null H1299 cell lines. The repression of CK1α expression significantly restored the tumor metastatic ability inhibited by NIFK downregulation in these p53-deficient cell lines. These results implicated the critical role of the NIFK-CK1α-β-catenin pathway in p53-deficient lung cancer. Previous studies have demonstrated that CK1α plays a role as a tumor suppressor in p53-inactivated cancer cells (*Elyada et al., 2011*; *Huart et al., 2009*; *Chen et al., 2005*). The loss of heterozygosity of CK1α results in highly invasive carcinoma in a p53-deficient mouse model (*Elyada et al., 2011*). Because p53 inactivation is a major pathogenic event in lung cancer (*Herbst et al., 2008*), further research is necessary to determine the role of NIFK-mediated CK1α expression in p53-deficient lung cancer.

Our study revealed that the molecular mechanism underlying NIFK-mediated CK1α downregulation in lung cancer is RUNX1-dependent at the transcriptional level. RUNX1 was originally recognized to display tumor-suppressive ability due to its role in acute myeloid leukemia tumorigenesis (*Miyoshi et al., 1991*; *Silva et al., 2003*). The loss of RUNX1 from intestinal epithelial cells significantly induced tumorigenesis in a conditional knockout mouse model (*Fijneman et al., 2012*), and the knockdown of RUNX1 in breast cancer cells resulted in hyperproliferation and abnormal morphogenesis (*Wang et al., 2011*; *Janes, 2011*). Cancer metastasis due to the inhibitory effect of NIFK on the binding of RUNX1 to the CK1α promoter region, as demonstrated by our study, represents a novel tumor progression mechanism that merits further investigation.

In conclusion, NIFK, a Ki-67-interacting protein, is first identified with clinical significance in lung cancer progression. NIFK enhances the metastatic ability of lung cancer cells via the Runx-1-dependent repression of CK1α expression and activates TCF/β–catenin signaling, thereby promoting metastasis in lung cancer. In addition, cell proliferation is positively regulated by NIFK via Ki-67. High NIFK expression correlates with poor prognosis and tumor metastasis in clinical lung cancer patients, suggesting that NIFK is an independent prognostic indicator and a promising therapeutic target.

## Materials and methods

### Patients

Tissue samples of non-small cell lung cancer, breast cancer, and colorectal cancer from patients were included to further analyze the clinicopathological role of NIFK. For non-small cell lung cancer, a total of 188 patients from Kaohsiung Medical University Hospital and National Taiwan University Hospital of Taiwan were included. Another tissue microarray using a Korean cohort of non-small cell lung cancer patients was purchased from SuperBioChips (SuperBioChips Laboratories, Seoul, Korea). For breast cancer, we examined samples from 84 patients from Kaohsiung Veterans General Hospital. For colorectal cancer, 62 patients were enrolled from Taipei Medical University Hospital. All cases were staged according to the cancer staging manual of the American Joint Committee on Cancer, and the histological cancer type was classified according to the World Health Organization classification. The tissues used were obtained with approval from the IRBs of Kaohsiung Medical

University Hospital (KMUH-IRB-20110286), National Taiwan University Hospital, Kaohsiung Veterans General Hospital (VGHKS12-CT9-057), and Taipei Medical University Hospital (IRB-99049). No informed consent was required because the data were analyzed anonymously

## Cell culture

PC9, PC13 and PC14 cells were developed at the National Cancer Center Hospital in Tokyo, Japan (*Lee et al., 1985*). The other nine human lung cancer cell lines were obtained from American Type Culture Collection (Manassas, VA, USA). Cell lines were purchased by Prof. Michael Hsiao at 2012 and 2013 with certificates of analysis including cell authentication by STR analysis and mycoplasma contamination test. The H1355, PC9, H358, H928, H520, H157, H661, and H460 cells were maintained in RPMI 1640 medium. The PC13, PC14, A549 and H1299 cells were maintained in DMEM. Each medium was supplemented with 10% fetal bovine serum, penicillin (100 units/ml), and streptomycin (100 μg/ml). The cells were incubated in a humidified atmosphere consisting of 95% air and 5% $CO_2$ at 37°C.

## Wound healing assay

The wound healing assay was assessed using culture inserts (Ibidi, Martinsried, Germany). The culture inserts were transferred to plates. The cells were seeded at a density of $2 \times 10^5$ cells/well and were allowed to attach. After incubation, the culture inserts were removed using sterile tweezers and washed with PBS. The plates were filled with culture medium supplemented with 2% serum to induce cell migration. The cells were photographed for quantification of closure of the exposed area. The denuded area closure was calculated by (Denuded distance $_{0\ h}$ − Denuded distance $_{Endpoint}$) / Denuded distance $_{0\ h}$.

## Animal study

All animal experiments were conducted in accordance with a protocol approved by the Academia Sinica Institutional Animal Care and Utilization Committee (IACUC, Protocol# 14-03-665). Age-matched male NSG mice (6 to 8 weeks of age) were used. To evaluate metastasis, $1 \times 10^6$ cells were resuspended in 0.1 ml of PBS and injected into the lateral tail vein (n=6). Metastatic lung nodules were counted and were further confirmed via HE staining using a dissecting microscope.

## Lentivirus-based shRNA production and infection

The lentiviral shRNA constructs were purchased from Thermo Scientific (Pittsburgh, PA, USA). Lentiviruses were produced via co-transfection of 293T cells with an shRNA-expressing plasmid, an envelope plasmid (pMD.G) and a packaging plasmid (pCMV-dR8.91) using calcium phosphate (Invitrogen, Carlsbad, CA, USA). The 293T cells were incubated for 18 hr, followed by replacement of the culture medium. The viral supernatants were harvested and titered at 48 and 72 hr post-transfection. The cell monolayers were infected with the indicated lentivirus in the presence of polybrene and were further selected using puromycin.

## TOP/FOP-Flash assay

The cells were co-transfected with the TOP-Flash (for TCF binding sites) or FOP-Flash (for mutated TCF binding sites) reporter plasmid (Millipore, Bedford, MA, USA) and pZsGreen (GFP) for 48 hr. The cells were harvested and lysed using a Promega luciferase assay kit (Madison, WI, USA) according to the manufacturer's instructions. The luminescence was measured using a luminometer. The TOP/FOP luminescence ratio was used as a measure of TCF/β-catenin transcriptional activity.

## Preparation of NIFK expression plasmid

NIFK was cloned from Beas2B cDNA using TAKARA DNA polymerase (Mountain View, CA, USA) according to the manufacture's instruction. The primer sequences designed were as follows: 5'-ACCCAAGCTGGCTAGCATGGCGACTTTTTCTGGCCCG-3' (sense) and 5'- TCAAGATCTAGAATTC TCACTGATTGCTGCTTCTTCG-3' (antisense). The PCR products were gel-purified, digested with NheI/EcoRI, and subcloned into lentiviral expression vector pLAS3W (RNAi Core, Academia Sinica, Taipei, Taiwan). The sequences were confirmed via DNA sequencing by Sequencing Core Facility, SIC, Academia Sinica.

## Western blot analysis

The cells were lysed at 4°C in RIPA buffer containing 50 mM Tris-HCl (pH 7.4), 150 mM NaCl, 1% Triton X-100, 0.25% sodium deoxycholate, 5 mM EDTA (pH 8.0), and 1 mM EGTA supplemented with protease and phosphatase inhibitors. After 20 min of lysis on ice, the cell debris was removed via microcentrifugation, followed by rapid freezing of the supernatants. The protein concentration was determined using the Bradford method. In our experiments, equivalent loads of 25–50 g of protein were electrophoresed using a SDS-polyacrylamide gel and then electrophoretically transferred from the gel to a PVDF membrane (Millipore, Bedford, MA, USA). After blocking with 5% non-fat milk, the membrane was incubated in specific primary antibodies (NIFK: Abcam, ab13880, 1:1000; Ki-67: Dako, Code M7240, 1:500; CK1α: Abcam, ab88079, 1:1000; non-phospho-β-catenin: Cell Signaling, #8814, 1:1000; phospho-β-catenin (Ser45): Cell Signaling, #9564, 1:1000; RUNX1: Cell Signaling, #4336, 1:1000) overnight at 4°C and subsequently incubated in a corresponding horseradish peroxidase-conjugated secondary antibody for 1 hr. The membranes were visualized using the ECL-Plus detection kit (PerkinElmer Life Sciences, Boston, MA, USA).

## Migration and invasion assay

The *in vitro* migration and invasion were assessed using Transwell assay (Millipore, Bedford, MA, USA). For invasion assay, transwell was additional pre-coated with 35 l of 3X diluted matrix matrigel (Bd Biosciences Pharmingen, San Diego, CA, USA) for 30 min. Cells of $2 \times 10^5$ in serum-free culture medium were added to the upper chamber of the device, and the lower chamber was filled with 10% FBS culture medium. After indicated hours of incubation, upper surface of the filter was carefully removed with a cotton swab. The filter was then fixed, stained and photographed. Cells of migration and invasion were quantified by counting the cells in three random fields per filter.

## Preparation of promoter constructs

The 1.3K bp *CSNK1A1* promoter fragment (Ensembl:ENSG00000113712) was cloned from genomic DNA of H1299 cells. The primer sequences designed were as follows: 5'- CCCCGGTACCCTGAC TTAAGATGATAGCAT-3' (sense) and 5'-TTTGCTAGCG CTGGGCCACT TGTTTCTCG-3' (antisense). The 1.1K bp *RUNX1* promoter was cloned from 293T cells using RUNX1_Promter1_F: 5'-AAAAGG TACCAAGCCAGTGGGGCCGGAAAA-3' and *RUNX1*_Promoter1_R: 5'-TGGGGCTAGCGGTTGTTTA TGAGGCCCAAA-3'. PCR was performed using TAKARA DNA polymerase (Mountain View, CA, USA) according to the manufacture's instruction. The PCR products were gel-purified, digested with KpnI/NheI, and subcloned into pGL4.20 firefly luciferase vector (Promega, Madison, WI, USA). The sequences of cloned promoter region were confirmed by DNA sequencing.

## Chromatin immunoprecipitation (ChIP) assay

ChIP assay was performed according to the manufacture's instruction (Abcam, Cambridge, MA, USA). Briefly, Cells were fixed with 1% formaldehyde and quenched by 1.25 M glycine. Cells were then lysed and immunoprecipitated overnight with antibodies against RUNX1 (Cell Signaling Technology, Danvers, MA, USA) or SRY (Santa Cruz, CA, USA). After immunoprecipitation, protein A beads were added for 1 hr to capture the immune complexes. The beads were washed, and the cross-link was further reversed. DNA was purified and analyzed by RT-PCR analysis. The primer sequences were as follows: RUNX1: 5'- ACTGAGGTTTTCAACAAGACCA -3' (sense) and 5'-ATCCCCCTGCCATCCTATGT-3' (antisense) with product size of 223 bp. SRY: 5'-ACCATGGAGTTTTCTTTCGTGA-3' (sense) and 5'-TGGTCTTGTTGAAAACCTCAGT-3' (antisense) with product size of 214 bp.

## Semi-quantitative RT-PCR and real-time PCR amplification analysis

Total cellular RNA was extracted by TRIzol reagent (Invitrogen, Carlsbad, CA, USA) in accordance with the manufacturer's instructions. One microgram of total RNA was reverse-transcribed using Advantage RT for PCR Kit (Clontech, Mountain View, CA, USA) at 42°C for 1 hr as described in the manufacturer's protocol. PCR conditions for rat leptin were 94°C for 5 min and 37 cycles at 94°C for 30 s, 56 C for 30 s and 72°C for 60 s, followed by a final extension step at 72°C for 5 min by Bio-Rad icycle (Bio-Rad). Primer sequences were as follows: human *CCND1*: 5'-GACCTTCGTTGCCCTCTGT-3' (sense) and 5'-TGAGGCGGTAGTAGGACAGG-3' (antisense) with product size of 180 bp; human *MMP7*: 5'-TGGGAACAGGCTCAGGACTAT-3' (sense) and 5'-CGTCCAGCGTTCATCCTCAT-3' (antisense) with

product size of 504 bp; human *TCF4*: 5'-CCGATGACGAGGGTGATGAG-3' (sense) and 5'-CCGAG-GACACCTTCTCTTCC-3' (antisense) with product size of 399 bp; human *CD44*: 5'-GGATCCACCC-CAACTCCATC-3' (sense) and 5'-AGGTCCTGCTTTCCTTCGTG-3' (antisense) with product size of 702 bp; human *MKI67IP* (NIFK): 5'-AGGTGGCGCAGGTTCGCAAG-3' (sense) and 5'-TGGTGTGGGGCCC TGGCTATC-3' (antisense) with product size of 632 bp; human *GAPDH*: 5'-GTCCACTGGCGTC TTCACCACC-3' (sense) and 5'-AGGCATTGCTGATGATCTTGAGGC-3' (antisense) with product size of 161 bp. For each combination of primers, the kinetics of PCR amplification was studied. The number of cycles corresponding to plateau was determined and PCR was performed at exponential range. PCR products were then electrophoresed through a 1% agarose gel and visualized by ethidium bromide staining in UV irradiation. The mRNA levels were also determined by real-time PCR with ABI StepOnePlus real-time PCR system according to the manufacturer's instructions. GAPDH was used as endogenous control. PCR reaction mixture contained the SYBR PCR master mix, 50 ng cDNA, and primers. Relative gene expression level that the amount of target were normalized to endogenous control gene was calculated using the comparative Ct method formula $E^{-\triangle\triangle Ct}$. The primer sequences were as follows:

RUNX1_F:5'-AGCCCCAACTTCCTCTGCTC-3'
RUNX1_R:5'-TCATCATTGCCAGCCATCAC-3'.

## Statistical analysis

Estimates of the survival rates were calculated using the Kaplan-Meier method and were compared using the log-rank test. The association between clinicopathological categorical variables and NIFK expression was analyzed using the chi-squared test. Student's *t*-test was used for other statistical analyses. All data are presented as the mean ± S.D. The p values at the following levels were considered to be significant: *$p<0.05$, **$p<0.01$, and ***$p<0.001$. All data was represented after at least three repeated experiments with a similar pattern.

## Acknowledgements

We thank Tracy Tsai for technical support and Linda Ho for assistance with the Affymetrix microarray experiment. This work was supported by the Thematic Project of Academia Sinica [AS-101-TP-B02] to Prof. Ming-Daw Tsai and Prof. Michael Hsiao, and by Ministry of Science and Technology [MOST 104-0210-01-09-02 and MOST 105-0210-01-13-01 to Michael Hsiao and MOST105-0210-01-12-01 to Ming-Daw Tsai].

## Additional information

### Funding

| Funder | Grant reference number | Author |
|---|---|---|
| Academia Sinica | AS-101-TP-B02 | Tsung-Chieh Lin<br>Chia-Yi Su<br>Pei-Yu Wu<br>Wen-An Pan<br>Ming-Daw Tsai<br>Michael Hsiao |

The funders had no role in study design, data collection and interpretation, or the decision to submit the work for publication.

### Author contributions

T-CLi, Conception and design, Acquisition of data, Analysis and interpretation of data, Drafting or revising the article, Contributed unpublished essential data or reagents; C-YS, Conception and design, Acquisition of data, Analysis and interpretation of data, Drafting or revising the article; P-YW, Analysis and interpretation of data, Drafting or revising the article, Contributed unpublished essential data or reagents; T-CLa, W-AP, Y-HJ, Y-CC, C-TY, C-LC, L-PG, H-TC, C-JY, M-SH, Contributed unpublished essential data or reagents; Y-PL, Analysis and interpretation of data, Drafting or

revising the article; Y-FL, JY-JS, Analysis and interpretation of data; M-DT, MH, Conception and design, Analysis and interpretation of data, Drafting or revising the article

### Author ORCIDs
Ming-Shyan Huang, ⓘ http://orcid.org/0000-0001-8180-2213
Michael Hsiao, ⓘ http://orcid.org/0000-0001-8529-9213

### Ethics
Human subjects: The tissues used were obtained with approval from the IRBs of Kaohsiung Medical University Hospital, National Taiwan University Hospital, Kaohsiung Veterans General Hospital, and Taipei Medical University Hospital.
Animal experimentation: All animal experiments were conducted in accordance with a protocol approved by the Academia Sinica Institutional Animal Care and Utilization Committee.

## Additional files

### Supplementary files
• Supplementary file 1. (A) Gene signatures regulated by lipofection-based NIFK overexpression in PC13 cell were listed. (B) Signature alternations after lentiviral-based NIFK overexpression in PC13 cell were listed.

### Major datasets
The following previously published dataset was used:

| Author(s) | Year | Dataset title | Dataset URL | Database, license, and accessibility information |
|---|---|---|---|---|
| Kohno T | 2011 | Gene expression data for pathological stage I-II lung adenocarcinomas | http://www.ncbi.nlm.nih.gov/geo/query/acc.cgi?acc=GSE31210 | Publicly available at the NCBI Gene Expression Omnibus (accession no. GSE31210) |

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
