## [Decision Letter]

Thank you for submitting your work entitled "NIFK promotes cancer progression via CK1α/β-catenin in metastasis and Ki-67-dependent cell proliferation" for consideration by *eLife*. Your article has been reviewed by two peer reviewers, one of whom, Narendra Wajapeyee, has agree to reveal his identity, and the evaluation has been overseen by Michael Green (Reviewing Editor) and Tony Hunter (Senior Editor).

3) In , the effect of CK1 upregulation and β-catenin downregulation is very modest. It is difficult to interpret the western blot data without quantification. Quantitative measurements are also lacking for .

4) In , the authors reported that NIFK overexpression leads to increase of total β-catenin. CK1a is known to phosphorylate β-catenin. The authors did not report levels of phosphorylated β-catenin, and need to do so.

5) In , RNA levels need to be quantified by QPCR.

---

## [Author Response]

*In this manuscript the authors report that NIFK, a Ki67 interacting protein, is associated with poor prognosis and metastasis in lung cancer. NIFK overexpression and shRNA-mediated knockdown was used to investigate the function of NIFK in human lung cancer cell lines.*

*NIKF is upregulated in lung cancer and regulates lung cancer proliferation and metastasis.*

*Mechanistically, the authors showed that NIFK downregulates CK1α, resulting in increased activity of β-catenin signaling. The authors also highlight Ki-67 dependent and independent mechanisms of NIKF action. NIKF mediated RUNX1 and CK1alpha regulation and their subsequent effect on WNT signaling has been well elucidated and adds to general understanding of the mechanism of NIKF action.*

*Essential revisions: 1) RUNX1 levels in microarray of PC13 cells overexpressing NIFK have not been indicated. They should be shown along with CK1a levels. Also, please highlight the WNT transcriptional target genes in the microarray.*

We thank reviewers for pointing out the question and giving us suggestion. We re-checked the microarray data of PC13 cells transiently overexpressing NIFK. RUNX1 level was not significantly downregulated (-1.14 in fold change). We therefore performed microarray analysis of PC13 cells stably overexpressing NIFK by lentiviral infection to confirm the axis. Please refer to the gene signatures listed in new , which showed significant CK1a (CSNK1A1) and RUNX1 downregulation upon NIFK overexpression (labeled in green). In addition, WNT signaling transcriptional target genes were labeled in red in the list. We further performed IPA analysis to identify the β-catenin transcriptional targets that are upregulated by NIFK. Please refer to the β-catenin transcriptional network in .

2) Mechanism of destabilization of RUNX1 mRNA has not been well studied, and needs further work. a. Does RUNX1 mRNA bind directly with NIKF?

*b. Do deletion/mutant constructs of NIKF that affect its binding activities have the same effect on RUNX1 mRNA levels.*

We thank reviewers for this important question. We analyzed the effects of NIFK deletion/mutant on RUNX1 mRNA levels. The result showed that RUNX1 mRNA was down regulated in A549 cells upon overexpression of wildtype NIFK. Furthermore, FHA, RRM domain deletion and T234A/T238A mutant all alleviated the repression effect toward RUNX1 at RNA level, suggesting that the molecular mechanism underlying the attenuated RUNX1 mRNA upon overexpression of NIFK is complicated and requires further investigation. Please refer to the result showed in .

*c. Is the decrease in RUNX1 binding to the CK1a promoter upon NIFK overexpression simply due to decreased RUNX1 protein or is there another mechanism? Could NIFK interact with RUNX1 protein? Here, RUNX1 protein levels need to be shown.*

Furthermore, the NIFK-mediated RUNX1 repression was reversely demonstrated in H1299 cells upon NIFK silencing. The silencing of NIFK promoted the level of RUNX1 protein that is presumably due to the attenuated repression of RUNX1 mRNA in the presence of NIFK, although a regulation at protein level in terms of stability may not be excluded. Please refer to the Western blot data in .

*d. A RUNX1 reporter construct can be used to show that RUNX1 mRNA transcription is not affected but its stability is affected, as a supplement to the actinomycin experiment.*

We thank the reviewers for the suggestion and we have performed the experiment accordingly. We have cloned RUNX1 promoter into pGL4.20 for reporter assay. Unexpectedly, we observed a significant decrease in promoter activity in A549 cells upon NIFK overexpression. This result together with our previous finding suggests the NIFK-mediated repression of RUNX1 mRNA is likely through a dual pathway mechanism. Please refer to the data showed in .

*3) In , the effect of CK1 upregulation and β-catenin downregulation is very modest. It is difficult to interpret the western blot data without quantification. Quantitative measurements are also lacking for .*

Thank you for this important suggestion. We have performed quantitative measurements for and Figure 6—figure supplement 1 using Gel-Pro Analyzer software. Band intensity was normalized to corresponding internal control (β-actin). The results collectively support our previous interpretation. Please refer to the quantitative measurement shown below and in the figures.

We thank the reviewers for the suggestion and we have performed the experiment accordingly. We have cloned RUNX1 promoter into pGL4.20 for reporter assay. Unexpectedly, we observed a significant decrease in promoter activity in A549 cells upon NIFK overexpression. This result together with our previous finding suggests the NIFK-mediated repression of RUNX1 mRNA is likely through a dual pathway mechanism. Please refer to the data showed in .

*4) In , the authors reported that NIFK overexpression leads to increase of total β-catenin. CK1a is known to phosphorylate β-catenin. The authors did not report levels of phosphorylated β-catenin, and need to do so.*

We thank the reviewers for the suggestion. We studied the phosphorylated β-catenin in A549 and PC13 cells. In agreement with the decreased level of CK1 α protein upon overexpression of NIFK (), our new results showed that the phosphorylation level of β-catenin were also decreased in response to overexpression of NIFK. Please refer to the result showed in .

*5) In , RNA levels need to be quantified by QPCR.*

We thank the reviewers for the suggestion. RNA levels were quantified by real-time PCR using specific primer targeting RUNX1 in A549 cells upon treatment of Actinomycin D. The trend is similar to what we previously showed in that overexpression of NIFK induced instability of RUNX1 mRNA. Please refer to the data showed in .

*6) Also add a correlation scatter plot for NIKF and CK1alpha from TCGA samples.*

We appreciate for the suggestion. We re-analyzed the correlation for NIFK and CK1alpha from the same TCGA lung cancer cohort database (genomic_TCGA_LUNG_exp_HiSeqV2_exon_clinical). However, the correlation is not as predicted. Another cohort with 513 cases of lung adenocarcinoma patients from TCGA (genomic_TCGA_LUAD_exp_HiSeqV2_percentile_clinical) was retrieved and analyzed. The result showed the statistically significant correlation among NIFK, RUNX1 and CK1alpha. Please refer to the new scatter plot in the manuscript ().

7) Are any of the observed effects of NIFK dependent on its interaction with Ki-67?

This question is critical for illustrating the role of Ki-67 in NIFK-mediated effects. In this study, we reported that NIFK-mediated cell proliferation likely depends on its interaction with Ki-67 in which (1) silencing of Ki-67, (2) truncation of FHA domain-interacting motif of NIFK, and (3) T234A/T238A mutation of NIFK all showed attenuated abilities in promotion of cell proliferation compared to NIFK wildtype overexpression group ().